# Expression of Estrogen Receptor Alpha and Evaluation of Histological Degeneration Scores in Fibroblasts of Hypertrophied Ligamentum Flavum: A Qualitative Study

**DOI:** 10.3390/biom11121752

**Published:** 2021-11-24

**Authors:** Christina C. Westhoff, Christian-Dominik Peterlein, Hanna Daniel, Juergen R. Paletta, Roland Moll, Annette Ramaswamy, Stefan Lakemeier

**Affiliations:** 1Institute of Pathology, Philipps University, and University Hospital Giessen and Marburg, 35033 Marburg, Germany; mollr@med.uni-marburg.de (R.M.); ramaswam@med.uni-marburg.de (A.R.); 2Center for Orthopedics and Trauma Surgery, Philipps University, and University Hospital Giessen and Marburg, 35033 Marburg, Germany; christian-dominik.peterlein@muehlenkreiskliniken.de (C.-D.P.); paletta@med.uni-marburg.de (J.R.P.); 3Institute of Medical Biometry and Epidemiology, Philipps University, 35037 Marburg, Germany; danielh@staff.uni-marburg.de; 4Department of Trauma Surgery, Orthopaedics and Plastic Surgery, University Medical Center Göttingen, 37075 Göttingen, Germany; stefan.lakemeier@googlemail.com

**Keywords:** ligamentum flavum, hypertrophy, lumbar spinal stenosis, estrogen receptor alpha, fibrosis, degeneration, proliferation, elastic fibers, collagen fibers, fibroblasts

## Abstract

The most common spinal disorder in elderly is lumbar spinal stenosis (LSS), resulting partly from ligamentum flavum (LF) hypertrophy. Its pathophysiology is not completely understood. The present study wants to elucidate the role of estrogen receptor α (ER α) in fibroblasts of hypertrophied LF. LF samples of 38 patients with LSS were obtained during spinal decompression. Twelve LF samples from patients with disk herniation served as controls. Hematoxylin & Eosin (H&E) and Elastica stains and immunohistochemistry for ER α were performed. The proportions of fibrosis, loss and/or degeneration of elastic fibers and proliferation of collagen fibers were assessed according to the scores of Sairyo and Okuda. Group differences in the ER α and Sairyo and Okuda scores between patients and controls, male and female sex and absence and presence of additional orthopedic diagnoses were assessed with the Mann–Whitney *U* test. There was a tendency towards higher expression of ER α in LF fibroblasts in the hypertrophy group (*p* = 0.065). The Sairyo and Okuda scores were more severe for the hypertrophy group but, in general, not statistically relevant. There was no statistically relevant correlation between the expression of ER α and sex (*p* = 0.326). ER α expression was higher in patients with osteochondrosis but not statistically significant (*p* = 0.113). In patients with scoliosis, ER α expression was significantly lower (*p* = 0.044). LF hypertrophy may be accompanied by a higher expression of ER α in fibroblasts. No difference in ER α expression was observed regarding sex. Further studies are needed to clarify the biological and clinical significance of these findings.

## 1. Introduction

One of the most common spinal disorder in the elderly is LSS [1], with LF hypertrophy being an important contributing factor [2]. However, its pathophysiology is not yet completely understood [3]. Ishimoto et al. demonstrated that the prevalence of LSS increases with age among women but not among men, pointing to a gender difference [4]. Estrogen deficiency is known as a major influence contributing to bone loss and osteoporosis after menopause [5], whereas Miyakoshi et al. reported an inverse relation between osteoporosis and spondylosis in postmenopausal women [6]. In the normal LF, elastic fibers are predominant, making up to 60–70% of the extracellular matrix [7]. The ratio of elastin to collagen fibers in the ligamentum flavum is 2:1, with the elastin fibers providing elasticity and the collagen fibers providing stiffness and stability [8]. With age, the elastin-to-collagen ratio decreases, diminishing the elasticity of the ligament [8]. In LF hypertrophy, the hypertrophied ligament shows an increase of collagen fibers, calcification, ossification, and chondrometaplasia [1,9,10], with fibroblasts presumably at the center of action [3]. There are a number of studies investigating the histopathological basis of LF hypertrophy and its correlation with clinical findings [1,2,7,8,9,10,11,12]. However, no generally approved score exists to quantify histopathological changes in LF pathology. Among the above-mentioned studies, the Sairyo and Okuda scores appear the most feasible, and especially, the Sairyo score was the most robust with regards to the interobserver reliability [1,9]. To date, no study exists to quantify ER α expression on whole and coherent LF patient materials with the relevant number of cases and clearly separating both sexes. The present study wants to elucidate the role of ER α in fibroblasts of hypertrophied LF on whole and coherent LF patient materials, as well as to investigate histopathological alterations of LF in our study population.

## 2. Materials and Methods

### 2.1. Patients

Fifty patients (29 male, 21 female) were prospectively included in our study. Study approval was obtained from the Ethics Committee of the Medical Faculty of the Philipps University of Marburg (study reference 191/09). Thirty-eight patients with LSS necessitating decompressive surgery comprised the study group, and LF specimens were harvested from the L4/L5 spinal level. The control group consisted of 12 patients with disk herniation at the L4/L5 level. None of the controls had signs of degeneration on preoperative magnetic resonance imaging (MRI). Care was taken to remove as little LF as possible.

### 2.2. Specimen Preparation

Tissues were fixed in 4% formalin solution, embedded in paraffin, cut at a thickness of 4 μm, and stained with H&E and Elastica van Gieson (EvG) stains, as established for routine histopathological diagnosis. Immunohistochemistry was performed using heat-induced antigen retrieval (employing citrate buffer pH 6.0) and the Leica Bond Polymer Refine Detection System with 3,3′-diaminobenzidine (DAB) as the chromogen (Leica Biosystems, Wetzlar, Germany). All immunostainings were run on an automated immunostaining apparatus (LEICA BOND-MAX, Leica Biosystems, Wetzlar, Germany). ER α was detected by the monoclonal mouse antibody NCL-ER6F11 (Novocastra, Leica Biosystems, Wetzlar, Germany).

### 2.3. Qualitative Assessment

The expression of ER α was semiquantitatively assessed as the nuclear staining of fibroblasts in LF according to the standardized procedures in breast cancer diagnostics by experienced senior pathologists (CCW and AR). The immunoreactive score (IRS) was calculated according to Remmele by multiplying the staining intensity (no reaction, 0; weak reaction, 1; moderate reaction, 2; strong reaction, 3) with the proportion of positive cells among all LF fibroblasts (0%, 0; <10%, 1; 10–50%, 2; 51–80%, 3; >80%, 4) [13]. The proportion of fibrosis and loss and/or degeneration of elastic fibers were assessed in H&E and EvG staining on the basis of the score of Sairyo [1,9]. Fibrosis was evaluated as the proportion of the entire LF tissue in percentage and as a score (grade 0, normal tissue without fibrosis; grade 1, fibrosis at ≤25% of the entire area; grade 2, fibrosis between 26 and 50% of the entire area; grade 3, fibrosis between 51 and 75% of the entire area; grade 4, fibrosis >75% of the entire area). The loss of elastic fibers was graded in percentage and using the same scoring system as with fibrosis. Furthermore, the loss of elastic fibers, degeneration of elastic fibers, and proliferation of collagen fibers were classified into mild, moderate, and severe, according to Okuda [9]. The pathologists evaluating the tissues were blinded to the clinical data of the study population. Two samples could not be evaluated due to a lack of ligamentum flavum tissue, one each among the control/LSS patient group and the male/female study participants, respectively.

### 2.4. Statistics

Differences with regards to sex, age, additional orthopedic diagnosis, ER α and proportion of fibrosis, loss and/or degeneration of elastic fibers, and proliferation of collagen fibers, according to Sairyo and Okuda between the groups of patients and controls, male and female study participants, and among LSS patients were evaluated with the Mann–Whitney *U*, Kruskal–Wallis-, *t*-test, or Fisher’s exact test using R [14] and compareGroups [15] where appropriate. Data will be deposited in data_UMR; accession numbers will be provided during review.

## 3. Results

The clinical characteristics, ER α expression data, and pathological findings of the control and the LSS patient groups are displayed in Table 1. While male and female study participants were equally represented in the LSS patient group, female patients were over-represented in the control group (*p* = 0.088). The LSS patients were significantly older than the study participants in the control group (*p* = 0.009), reflecting the typical age of onset for lumbar spinal stenosis (LSS patient group) and disk herniation (control group).

Immunohistochemical staining for ER α in the nuclei of LF fibroblasts could be demonstrated for both sexes, along with a partly impressive staining intensity for the male probes (Figure 1c,f). There was a tendency towards a higher expression of ER α in the LSS patient group (*p* = 0.065). In the LF specimens, the Sairyo and Okuda scores were in general more severe for the LSS patient group but, with regards to the differences between the control and LSS patient groups, not statistically relevant. Figure 1 displays two cases exemplarily with moderate amounts of fibrosis (a) and a medium loss of elastic fibers (b), as well as low amounts of fibrosis (d) and a moderate loss of elastic fibers (e) from patients within the control and LSS patient groups. Figure 2 exemplifies two LSS patient cases with more drastically different degrees of fibrosis and loss of elastic fibers.

Table 2 illustrates the differences between the male and female study participants. There was no statistically relevant correlation between age (*p* = 0.127), additional orthopedic diagnosis (*p* = 0.365), or ER α (*p*= 0.326) and sex. Furthermore, in both sexes, the same range of ER α expression was found (minimum IRS 0, maximum IRS 8), especially among LSS patients (Appendix A). No statistically significant difference was found in the Sairyo and Okuda scores between the male and female study participants.

Among the affected study participants, i.e., the LSS patient group, the ER α expression was higher in patients with the additional diagnosis of osteochondrosis but not statistically significantly correlated (*p* = 0.113) (Table 3). In patients with the additional diagnosis of structural instability, no statistically significant difference was seen in ER α expression (*p* = 0.932). However, in patients with scoliosis, the ER α expression was significantly lower (*p* = 0.044). When evaluating the absolute fibrosis and loss of elastic fibers on the basis of the score of Sairyo among the LSS patient group (Table 3), there was statistically significantly less fibrosis in patients with osteochondrosis (*p* = 0.04), but there was no difference for structural instability (*p* = 0.124) or scoliosis (*p* = 0.125). There was also a trend for less of a loss of elastic fibers in patients with osteochondrosis (*p* = 0.076). No statistically significant effect was seen for the loss of elastic fibers in patients with structural instability (*p* = 0.308) or scoliosis (*p* = 0.765). The scores of fibrosis and the loss of elastic fibers according to Sairyo and the scores according to Okuda showed no statistically significant effects with regards to additional diagnoses (Appendix A).

## 4. Discussion

Hypertrophy of the LF is an important contributing factor in LSS [2], being one of the most common spinal disorders in elderly patients [1]. However, the exact mechanism of fibrosis formation remains unknown despite numerous studies [3]. Fibroblasts are the central mediators of pathological fibrotic accumulation of the extracellular matrix and constitute a suitable therapeutic target [16], having been linked to the pathogenesis of LF hypertrophy [3,17]. Ishimoto et al. demonstrated that the prevalence of LSS increases with age among women but not among men, pointing to a gender difference [4]. Estrogen deficiency is known as a major influence contributing to bone loss and osteoporosis after menopause [5], whereas Miyakoshi et al. reported an inverse relation between osteoporosis and spondylosis in postmenopausal women [6]. Therefore, we wanted to elucidate the role of the estrogen receptor (ER) in fibroblasts of hypertrophied LF.

In previous studies, expression of the ER was demonstrated on different tendons and ligaments via immunohistochemistry or reverse transcription-polymerase chain reaction (RT-PCR) [18,19,20,21,22], including male study subjects. However, to date, only one study examined ER on LF [5]. In that study, Chen et al. demonstrated a similar staining density for ER β in a female and male probe each on alcohol-fixed cytospins, while ER α was only identified on the female LF cells at a reduced density [5]. However, they did not use whole and coherent LF tissues and tested only one probe per sex by immunocytochemistry on cytospins of isolated LF cells. In contrast, we examined paraffin-embedded coherent tissue probes from 50 persons for ER α in a semiquantitative manner according to the standardized procedures in breast cancer diagnostics. We found no statistically significant difference between male and female probes (Table 2). In fact, identical medians and minimums or maximums could be demonstrated for both sexes, along with a partly impressive staining intensity for male probes (Figure 1f). Therefore, our study is the first published study to examine ER α on patient materials with a relevant number of cases.

In our study population, we found a tendency toward higher ER α expression in LF fibroblasts of LSS patients among both sexes, accompanied by a higher variation among LSS patients (Appendix A). ER α has been shown to regulate vascular endothelial growth factor (VEGF) expression [23], while VEGF is known for mediating angiogenesis [23,24]. Additionally, phytoestrogens promote angiogenesis via ER α [25]. For LF, hypertrophy has been shown to be associated with increased VEGF expression [26], and VEGF-mediated angiogenesis might be a critical step in LF hypertrophy [27]. Furthermore, in an in vitro study, estradiol upregulated the expression of matrix metalloproteinase 13 (MMP-13) via the PI3K pathway and contributed to the homeostasis of the extracellular matrix in LF [5]. MMP-13 has been reported as being more highly expressed in patients with LSS and being localized in LF fibroblasts [8]. Our findings also emphasize the potential involvement of estradiol/ER α in the pathogenesis of LSS, though a larger patient sample may be needed for statistical significance.

Since the LSS patients were much older than the study participants of the control group, it should be considered whether age alone could lead to a higher ER α expression. However, in previous studies on ER α in mesenchymal cells, no such correlation was found. In smooth muscle cell nuclei of the uterosacral ligament, the ER was detected immunohistochemically in all 25 patients studied regardless of variations in patient ages, which ranged from 34 to 68 years [21]. In mesenchymal cells of the anterior cruciate ligament in patients with osteoarthritis ranging in ages from 57 to 78 years, nuclear ER immunoreactivity was detected in eight out of eleven cases, with the negative cases showing even a higher median age as compared to the positive cases [28]. Thus, we do not believe that the increase in ER α expression in LSS patients vs. control participants is related to the higher ages of the patients.

Among LSS patients, the ER α expression was higher with the additional diagnosis of osteochondrosis, with a tendency toward statistical significance, whereas ER α expression was significantly lower in patients with the additional diagnosis of scoliosis. To date, no correlation regarding estrogen/ER and osteochondrosis has been found in the pertinent literature. With respect to scoliosis, ER α is considered a candidate gene for idiopathic scoliosis [29]. Furthermore, adolescent idiopathic scoliosis is associated with a higher estrogen serum concentration, an unusual cellular response to estrogen, late age at menarche, and gene polymorphism of ER [30]. Though our results (lower ER α expression in LSS patients with scoliosis) are contradictory in this context, the patient population under scrutiny corresponded to the typical population of LSS, with a median age of 67.5 years, contrasting the typical age of patients with (adolescent) idiopathic scoliosis. Additionally, only one female patient was found in our study population with scoliosis (Table 2), possibly contributing to the lower ER α expression. A larger study cohort may help to further clarify this correlation.

There have been a number of studies investigating the histopathological basis of LF hypertrophy and its correlation with clinical findings. However, no generally approved score exists to quantify histopathological changes in LF pathology. Therefore, we used the scores proposed by Sairyo et al. [1] and Okuda et al. [9] for quantification. While Sairyo et al. demonstrated moderate-to-strong positive linear correlations between fibrosis/loss of elastic fibers and thickness of the LF, they did not perform extensive statistical analyses. Additionally, a histological analysis was only done for 20 patients, while we examined materials from almost double that number. Furthermore, Sairyo et al. did not include a control group and excluded patients with fractures, scoliosis, and spondylolisthesis, thereby obtaining a homogenous, albeit clinically unrealistic, study population. Additionally, the mean age of their patients qualifying for a histological analysis was 56.2 years and ranged between 34 and 75, being relatively young for patients with LSS.

Okuda et al. found no significant relationship between the loss of elastic fibers, degeneration of elastic fibers, or proliferation of collagen fibers and clinical symptoms or image findings. Although they examined materials from 50 patients, they did not include a control group without degenerative changes but analyzed only patients with LSS or spondylolisthesis.

In a gender-matched case–control study of 30 patients with LSS or disc herniation, respectively, Park et al. identified a statistically significant correlation of elastin degradation and fibrosis with LSS [8] while grading the elastin degradation and fibrosis according to Sairyo et al.

In our study population—being slightly larger with respect to LSS patients than Park’s cohort—the scores according to Sairyo and Okuda, respectively, were, in general, more severe for the LSS patient group but, with regard to the differences between the control and LSS patient groups, not statistically relevant.

Sairyo et al. demonstrated increased fibrosis and loss of elastic fibers, particularly in the dorsal portion of the LF [1]. In Park’s study, an effort was made to obtain the entire layer of the central portion of the LF, thereby obviously gaining more materials and enough substances to explore at least 10 high-power fields.

While our results on the histopathological changes in LF hypertrophy are in-line with previous findings and promising, further studies are warranted, with preferably more patient samples examined and LF tissues gained and processed in an oriented manner, e.g., for the differentiation of dural and dorsal LF. Further, larger studies may aid in elucidating the potential clinical consequences of our findings.

Concerning the small number of control samples in our study (*n* = 12) compared with the hypertrophy group, we only used LF samples from patients with an indication for discectomy without visible signs of degenerative lumbar spine disease, being a rare clinical situation. Additionally, only a few male control samples were available (*n* = 2), representing a potential bias. However, care was taken to recruit an equivalent number of male and female patients (*n* = 19 each).

## 5. Conclusions

Our study is the first published to examine ER α on LSS patient materials with a relevant number of cases and clearly indicating both sexes. LF hypertrophy may be accompanied by a higher expression of ER α. More severe fibrosis, loss of elastic fibers, and the proliferation of collagen fibers, according to Sairyo and Okuda, were observed in the hypertrophy group. No statistically relevant correlation was seen between ER α and sex. Among the LSS patients, ER α expression was higher with the additional diagnosis of osteochondrosis, with a tendency toward statistical significance, whereas the ER α expression was significantly lower in LSS patients with the additional diagnosis of scoliosis. Further, larger studies with some technical modifications in material processing may help to clarify the biological and clinical significances of these findings.

## Figures and Tables

**Figure 1 biomolecules-11-01752-f001:**
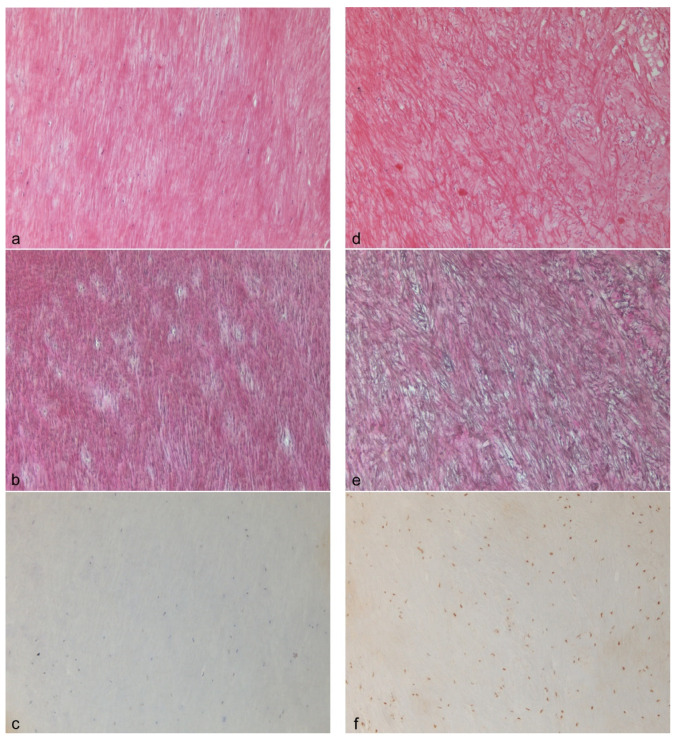
Histological and immunohistochemical profiles of two cases with low estrogen receptor α expression (**a**–**c**) and high estrogen receptor α expression (**d**–**f**). The first case displays ligamentum flavum tissue with a moderate amount of fibrosis ((**a**) H&E, 100×) and medium loss of elastic fibers ((**b**) EvG, 100×). No estrogen receptor α positivity was observed in the nuclei of the tissue (immunoreactive score (IRS) 0/12) ((**c**) 100×). The patient is an 81-year-old female patient belonging to the control group. The second case depicts ligamentum flavum tissue with a low amount of fibrosis ((**d**) H&E, 100×) and a moderate loss of elastic fibers ((**e**) EvG, 100×). Relatively high nuclear estrogen receptor α positivity in LF fibrocytes was observed in this example (IRS 8/12) ((**f**) 100×). The patient is a 63-year-old male diagnosed with lumbar spinal stenosis and osteochondrosis in the LSS patient group.

**Figure 2 biomolecules-11-01752-f002:**
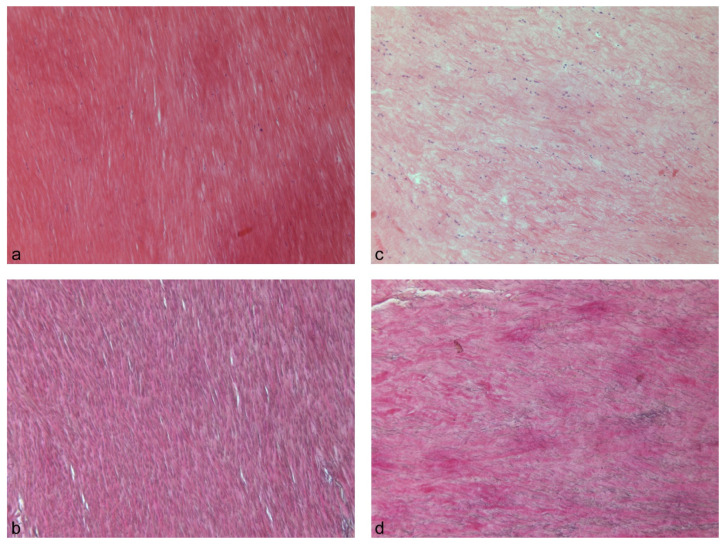
Histological profile of two LSS patient cases illustrating different degrees of fibrosis (**a**,**c**) and loss of elastic fibers (**b**,**d**). The first case presents with a mild degree of fibrosis ((**a**) H&E, 100×) and a mild loss of elastic fibers ((**b**) EvG, 100×). The patient is a 68-year-old male patient diagnosed with lumbar spinal stenosis and osteochondrosis in the LSS patient group. The second example exhibits a high amount of fibrosis ((**c**) H&E, 100×), as well as a moderate-to-severe loss of elastic fibers ((**d**) EvG, 100×). The patient is a 52-year-old female patient diagnosed with lumbar spinal stenosis and structural instability in the LSS patient group.

**Table 1 biomolecules-11-01752-t001:** Patient characteristics of the control and patient groups regarding clinical data, ER α expression and histopathology.

	Control Group (*N* = 12)	LSS Patient Group (*N* = 38)	*p* Overall
**Sex**			0.088
Male	2 (16.7%)	19 (50.0%)	
Female	10 (83.3%)	19 (50.0%)	
Age	44.5 (20.0; 83.0)	67.5 (50.0; 85.0)	0.009
**Additional orthopedic diagnosis**			.
Osteochondrosis	0 (0.00%)	23 (60.5%)	
Structural instability	0 (0.00%)	10 (26.3%)	
Scoliosis	0 (0.00%)	5 (13.1%)	
Estrogen receptor α (IRS)	2.00 (0.00; 3.00)	3.00 (0.00; 8.00)	0.065
Sairyo fibrosis absolute (% of total area)	30.0 (20.0; 50.0)	35.0 (15.0; 80.0)	0.494
Sairyo fibrosis Score (min. 0, max. 4)	2.00 (1.00; 2.00)	2.00 (1.00; 4.00)	0.631
Sairyo loss of elastic fibers absolute (% of total area)	25.0 (15.0; 45.0)	30.0 (10.0; 80.0)	0.464
Sairyo loss of elastic fibers score (min. 0, max. 4)	1.00 (1.00; 2.00)	2.00 (1.00; 4.00)	0.684
**Okuda degeneration of elastic fibers**			1.000
Mild	7 (63.6%)	22 (59.5%)	
Moderate	4 (36.4%)	13 (35.1%)	
Severe	0 (0.00%)	2 (5.41%)	
**Okuda proliferation of collagen fibers**			1.000
Mild	3 (27.3%)	12 (32.4%)	
Moderate	8 (72.7%)	23 (62.2%)	
Severe	0 (0.00%)	2 (5.41%)	
**Okuda loss of elastic fibers**			1.000
Mild	6 (54.5%)	18 (48.6%)	
Moderate	5 (45.5%)	17 (45.9%)	
Severe	0 (0.00%)	2 (5.41%)	

Continuous variables are described with counts and percentages in parentheses, and nominal variables are defined by the median, as well as the minimum and maximum in parentheses. *p*-values between the control and patient groups were calculated with the Mann–Whitney *U* test. One case each of the control and the LSS patient group could not be evaluated for ER α, Sairyo, and Okuda due to lack of materials.

**Table 2 biomolecules-11-01752-t002:** Comparing characteristics between the male and female participants regarding the group, clinical data, ER α expression, and histopathology.

	Male (*N* = 21)	Female (*N* = 29)	*p* Overall
**Control vs. LSS patient group**			0.088
Control group	2 (9.52%)	10 (34.5%)	
LSS patient group	19 (90.5%)	19 (65.5%)	
Age	71.0 (37.0; 83.0)	62.0 (20.0; 85.0)	0.127
**Additional orthopedic diagnosis**			0.365
Osteochondrosis	10 (62.5%)	13 (44.8%)	
Structural instability	4 (19.0%)	6 (20.7%)	
Scoliosis	4 (19.0%)	1 (0.3%)	
Estrogen receptor α (IRS)	2.00 (0.00; 8.00)	2.00 (0.00; 8.00)	0.326
Sairyo Fibrosis absolute (% of total area)	35.0 (15.0; 80.0)	30.0 (20.0; 60.0)	0.351
Sairyo Fibrosis Score (min. 0, max. 4)	2.00 (1.00; 4.00)	2.00 (1.00; 3.00)	0.172
Sairyo Loss of Elastic fibers absolute (% of total area)	30.0 (10.0; 80.0)	25.0 (10.0; 50.0)	0.403
Sairyo Loss of Elastic fibers Score (min. 0, max. 4)	1.50 (1.00; 4.00)	1.50 (1.00; 2.00)	0.867
**Okuda degeneration of elastic fibers**			1.000
Mild	12 (60.0%)	17 (60.7%)	
Moderate	7 (35.0%)	10 (35.7%)	
Severe	1 (5.00%)	1 (3.57%)	
**Okuda proliferation of collagen fibers**			0.768
Mild	5 (25.0%)	10 (35.7%)	
Moderate	14 (70.0%)	17 (60.7%)	
Severe	1 (5.00%)	1 (3.57%)	
**Okuda loss of elastic fibers**			0.537
Mild	8 (40.0%)	16 (57.1%)	
Moderate	11 (55.0%)	11 (39.3%)	
Severe	1 (5.00%)	1 (3.57%)	

Continuous variables are described with counts and percentages in parentheses, and nominal variables are defined by the median, as well as the minimum and maximum, in parentheses. *p*-values between male and female study participants are calculated with the Mann–Whitney *U* test. One case each of male and female study participants could not be evaluated for ER α, Sairyo, and Okuda due to lack of materials.

**Table 3 biomolecules-11-01752-t003:** Influence of additional orthopedic diagnoses on the expression of ER α and histopathology among LSS patients.

**Osteochondrosis**	**No (*N* = 15)**	**Yes (*N* = 23)**	***p* Overall**
Estrogen receptor α (IRS)	2.50 (1.83)	3.70 (2.64)	0.113
Sairyo Fibrosis absolute (% of total area)	42.1 (17.0)	31.2 (9.27)	0.040
Sairyo Loss of Elastic fibers absolute (% of total area)	35.7 (17.0)	26.3 (10.2)	0.076
**Structural instability**	**No (*N* = 28)**	**Yes (*N* = 10)**	***p* Overall**
Estrogen receptor α (IRS)	3.22 (2.45)	3.30 (2.41)	0.932
Sairyo Fibrosis absolute (% of total area)	33.1 (12.9)	41.5 (14.3)	0.124
Sairyo Loss of Elastic fibers absolute (% of total area)	28.5 (14.3)	33.5 (12.3)	0.308
**Scoliosis**	**No (*N* = 33)**	**Yes (*N* = 5)**	***p* Overall**
Estrogen receptor α (IRS)	3.45 (2.44)	1.50 (1.29)	0.044
Sairyo Fibrosis absolute (% of total area)	34.8 (14.3)	40.0 (4.08)	0.125
Sairyo Loss of Elastic fibers absolute (% of total area)	29.7 (14.4)	31.2 (8.54)	0.765

Variable values equal the variable mean; the numbers in parentheses correspond to the standard deviation. *p*-values between the absence and presence of the respective additional orthopedic diagnoses were calculated with the *t*-test. One case of LSS patients could not be evaluated for ER α and Sairyo due to a lack of materials.

## Data Availability

The data presented in this study are available upon request from the corresponding author. The data are not publicly available due to the legal and ethical restrictions of a research trial.

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
