# Peer review of "Expression of Estrogen Receptor Alpha and Evaluation of Histological Degeneration Scores in Fibroblasts of Hypertrophied Ligamentum Flavum: A Qualitative Study"

_biomolecules, 2021, doi:10.3390/biom11121752_

Round 1
Reviewer 1 Report
Material Methods
In the introduction, the patient's age is mentioned, but only the level of surgery is mentioned.
Isn't the site where ER is measured differently for the dural and dorsal sides?
Describe in detail the scores of Sairyo and Okuda. It is unkind.
The basic content is strong; the Discussion should describe how the current study will be helpful in clinical practice.
Reviewer 2 Report
The submitted manuscript attempts to perform a qualitative study investigating the expression of ECG receptor alpha correlated to degenerative scores in fibroblasts of hypertrophied ligament flavum.
The manuscript is well-written and flows logically.
While the study is an interesting one as it attempts to correlate ER alpha receptor expression to the pathogenesis of LF hypertrophy, there are some concerns regarding the manuscript and data analysis which I have summarised below.
- The methods section should be written with subheadings describing human ethics and participants, methods for histological preservation and preparation, immunohistochemistry, histological and immunohistochemical qualitative assessment and statistics. These are quite standard with respect to manuscript presentation and makes easy reading.
- What is IRS? Page 2, line 75.
- There is insufficient detail describing the quantification or quantitative assessment of the antibody stain. Was this done by a software? If so, was the intensity density used for perhaps another parameter to quantify the staining per cell?
- While the study include more participants in previous studies in the investigation of exigent receptor expression in LSS, there is a clear sex bias in patients recruited between control groups and LSS patients. This was not discussed or directly addressed anywhere in the manuscript except that understandably the recruitment of control patients is inherently difficult owing to the nature of this tissue sample. However a sex bias cannot be discounted and therefore a sub analysis should be performed in the female cohort and the limitation could be described in the discussion with respect to the expression of ER receptors in control males compare to male counterparts in the LS group. The difference in ER expression regarding sexes cannot be reasonably concluded given there are only 2 male participants in the control group compared to 19 LSS patients. This limitation should be clearly stated. The concluding statement (page 9, line 260) should hence be revised as I don't agree there is a clear separation of both sexes.
- Moreover, there is a large discrepancy with respect to age between control and LSS groups, although difference in age between sexes irrespective of groups is smaller (9 years). Is the expression of ER receptors known to diverging with age?
- Page 9, line 264: It's not clear if the authors are referring to LSS patients or control patients, or more problematic both? In the latter case, the small number of males in the control group makes this statement difficult to defend.
- As a minor point regarding presentation of figures, it will make easier reading for labels to be positioned over representative immuno-histochemical microphotographs.
Round 2
Reviewer 1 Report
I have no more comments.
Author Response
Thank you for participating in the peer review of this manuscript